# Doped Graphene Quantum Dots as Biocompatible Radical Scavenging Agents

**DOI:** 10.3390/antiox12081536

**Published:** 2023-07-31

**Authors:** Adam Bhaloo, Steven Nguyen, Bong Han Lee, Alina Valimukhametova, Roberto Gonzalez-Rodriguez, Olivia Sottile, Abby Dorsky, Anton V. Naumov

**Affiliations:** 1Department of Physics and Astronomy, Texas Christian University, Fort Worth, TX 76129, USA; bhalooad@seas.upenn.edu (A.B.); s.nguyen@yale.edu (S.N.); bong.lee@tcu.edu (B.H.L.); a.valimukhametova@tcu.edu (A.V.); olivia.sottile@tcu.edu (O.S.); a.dorsky@tcu.edu (A.D.); 2Department of Chemistry and Biochemistry, Texas Christian University, Fort Worth, TX 76129, USA; roberto.gonzalezrodriguez@unt.edu

**Keywords:** graphene quantum dots, antioxidants, heteroatom doping, biocompatibility, reactive oxygen species

## Abstract

Oxidative stress is proven to be a leading factor in a multitude of adverse conditions, from Alzheimer’s disease to cancer. Thus, developing effective radical scavenging agents to eliminate reactive oxygen species (ROS) driving many oxidative processes has become critical. In addition to conventional antioxidants, nanoscale structures and metal–organic complexes have recently shown promising potential for radical scavenging. To design an optimal nanoscale ROS scavenging agent, we have synthesized ten types of biocompatible graphene quantum dots (GQDs) augmented with various metal dopants. The radical scavenging abilities of these novel metal-doped GQD structures were, for the first time, assessed via the DPPH, KMnO_4_, and RHB (Rhodamine B protectant) assays. While all metal-doped GQDs consistently demonstrate antioxidant properties higher than the undoped cores, aluminum-doped GQDs exhibit 60–95% radical scavenging ability of ascorbic acid positive control. Tm-doped GQDs match the radical scavenging properties of ascorbic acid in the KMnO_4_ assay. All doped GQD structures possess fluorescence imaging capabilities that enable their tracking in vitro, ensuring their successful cellular internalization. Given such multifunctionality, biocompatible doped GQD antioxidants can become prospective candidates for multimodal therapeutics, including the reduction of ROS with concomitant imaging and therapeutic delivery to cancer tumors.

## 1. Introduction

Oxidative stress is an imbalance between reactive oxygen species (ROS) such as superoxide anion (O_2_^•−^), hydrogen peroxide (H_2_O_2_), and hydroxyl radicals (^•^OH) and biological defense mechanisms capable of the reduction and detoxification of these harmful moieties. Reactive oxygen species are naturally produced by the body as a byproduct of cellular metabolism and play important roles in cellular signaling pathways. However, the excess or inability of the body to circumvent those can be associated with substantial oxidative stress-related damage in countless diseases, including diabetes [1], Alzheimer’s disease [2,3,4,5,6,7], rheumatoid arthritis [8,9], and cancer [10].

ROS can be generated during mitochondrial oxidative metabolism and cellular responses to bacterial infections. Excessive amounts of ROS in cells can also arise from UV radiation [11,12], pollution [11], and ingestion of several chemical compounds, including pesticides [13,14]. Mammalian cells maintain a balanced amount of ROS as a vital mechanism for homeostasis and cellular proliferation. However, an excess of ROS can lead to electron imbalance, causing continuous electron transfer reactions, eventually manifesting in oxidative stress. Long-term exposure to oxidative stress can have detrimental effects on biological systems, including damage to cellular components such as lipids, proteins, and DNA [15], leading to cancer-generating mutations. Enzymatic reactions can also generate ROS in cells, including hydroxyl radicals (HO•), superoxide (O_2_^•−^), lipid hydroperoxides, and reactive nonradical compounds such as chloramines (RNHCl), hypochlorous acid (HOCl), and ozone (O_3_) [16]. ROS can overload the brain’s calcium supply and initiate unwanted nerve responses throughout the body. High concentrations of calcium, instigated by reactive oxygen species (ROS), trigger enzymes responsible for the degradation of cellular components. This initiates a cascade of events that ultimately leads to cell death [17,18,19,20,21]. The levels of ROS and antioxidant defenses also play a significant role in the development and progression of Alzheimer’s disease by creating local variations of oxidative stress. Alzheimer’s is caused by the accumulation of amyloid beta and tau protein aggregates, which both contribute to the formation of plaques in the brain. ROS can trigger the production of these proteins by activating the enzymes and signaling pathways involved in breaking down their precursors [22]. Mitochondrial oxidative stress may be an important contributor to the pathogenesis of Alzheimer’s disease due to higher levels of oxidative DNA damage in mitochondria of the frontal, parietal, and temporal lobes [23,24]. Mitochondrial dysfunction, whether temporary or long-lasting, can cause a decrease in ATP levels, boost ROS generation, and initiate apoptosis, which can ultimately lead to neurodegeneration [25]. Considering a multitude of adverse effects, there is a critical need to manage ROS concentrations. Radical scavenging agents, or antioxidants, can alleviate oxidative stress by donating extra electrons to quench the reaction and prevent the aforementioned detrimental effects. The development of effective and biocompatible antioxidants is, therefore, vital to addressing a variety of complex health conditions [1,5,6,7,8,26,27,28,29,30,31,32,33,34,35,36,37,38]. Currently, vitamin C [26,39], vitamin E [26,35,40,41,42,43,44], and beta carotene [45,46] are clinically proven antioxidant supplements capable of bolstering the body’s defense against ROS, thereby alleviating oxidative stress. However, those supplements provide only a moderate effect that is not focused on the disease site, where oxidative stress is manifested. A more targeted and multifaceted therapeutic approach can be facilitated by nanomaterials-driven antioxidant therapies.

Recent studies have shown that carbon nanomaterials, including carbon nanotubes [47], carbon dots [48], and graphene [49], can scavenge ROS by an electron transfer from their electron-rich platform or via hydrogen donation. In addition to being prospective antioxidants, nanocarbons can also serve as multifunctional agents for a variety of biomedical applications. For instance, functionalized carbon nanotubes have already been utilized for targeted delivery of small molecule drug and gene therapeutics in vitro and in vivo [50,51], as well as for biological imaging [52,53,54,55]. However, their further clinical applications are delayed due to the apparent toxicity of some CNT formulations [56,57]. With the continued development of multifunctional graphene derivatives, graphene quantum dots and carbon dots can move to the forefront of biomedical applications. These nanomaterials can also scavenge ROS with the same efficiency [58,59] while having higher biocompatibility than their counterparts.

Carbon dots present carbon-rich yet amorphous structures with a variety of different nanoscale sizes. Graphene quantum dots (GQDs) usually come as a few nanometer-sized nanoparticles composed of several graphene layers. Some GQD structures possess functional groups on their surface [60], rendering them water-soluble. Many of those synthesized from biological molecular precursors also demonstrate exceptionally high biocompatibility [61,62]. Quantum confinement, as well as functionalization and doping of GQDs, can manifest in unique optical properties, including visible and near-infrared fluorescence suitable for biomedical imaging [63,64]. In addition to being prospective imaging agents, GQDs can be synthesized from different molecular compounds, and their structure can be tailored to a particular application, including drug delivery [65,66,67,68], photothermal therapy [68,69,70], imaging [64,69,71], and biomarker sensing [71,72]. GQDs can be targeted to a specific disease site [73], which can help focus their potential antioxidant effects.

Recent in vitro and in vivo studies show that GQDs and carbon dots can protect cells and tissues from oxidative stress-induced damage. Both GQDs [58,59] and carbon dots [48,74,75,76,77,78] can exhibit antioxidant activity and effectively scavenge free radicals, such as hydroxyl radicals and superoxide anions. Doping GQD structures can further enhance their antioxidant properties due to the increased electron donation or reactivity introduced by the dopants [79]. Heteroatom doping can vary both the physical and electronic structure of the GQDs, adding ultrasound and MRI contrast capabilities [80,81]. Furthermore, doping can alter the photoluminescence capabilities of the GQDs, enabling biomedical imaging in near-infrared [82]. Our work aims to develop GQD antioxidants with such doping-enabled multimodal capabilities.

In order to enhance the antioxidant properties of the GQDs while retaining their image-tracking capabilities, we synthesize multiple biocompatible metal-doped structures, including Ag-, Al-, Ce-, Fe-, Ho-, MoS_2_-, Nd-, and Tm-GQD via a bottom-up hydrothermal reaction with a single common glucosamine precursor. The effects of metal doping are investigated in comparison to top-down-synthesized undoped GQDs via three major radical scavenging assays, DPPH, KMnO_4_, and dye protectant assay, while their imaging capabilities, as well as biocompatibility, are evaluated in vitro. Previous studies have shown that undoped GQDs have the potential to serve as antioxidants and that doping GQDs, such as chlorine and phosphorus, can reduce free radicals almost 60% more effectively than undoped GQDs [79,83,84,85]. The ultimate goal of this work is to develop novel doped GQD platforms that will not only enable effective ROS scavenging but will also be capable of image tracking and drug delivery for a variety of clinical applications.

## 2. Materials and Methods

### 2.1. Synthesis

#### 2.1.1. Bottom-Up Approach

Ag-, Al-, Ce-, Fe-, Ho-, MoS_2_-, N-, Nd-, and Tm-GQDs were synthesized using a hydrothermal bottom-up approach in a commercially available microwave oven (HB-P90D23AP-ST, Hamilton Beach, Glen Allen, VA, USA) following the procedure described previously [80]. Briefly, aqueous solutions consisting of 4 g of glucosamine hydrochloride (glucosamine hydrochloride, Lot #3510840, Millipore Sigma, Burlington, MA, USA) with the addition of 10 mL of silver nanoparticles (NP), or 2 mmol of CeO_2_, or 1 mL of iron oxide NP (Iron(III) oxide >99%, CAS #1309-37-1, Chemcraft Ltd., Kaliningrad, Russia), or 4 mmol of Ho(NO_3_)_3_⋅5H_2_O (Holmium(III) nitrate >99%, Chemcraft Ltd., CAS #14483-18-2), or 0.3 mmol of MoS_2_ (molybdenum disulfide >99%, CAS #1317-33-5, LoudWolf, Dublin, CA, USA) and 0.15 mmol of Sodium n-dodecyl sulfate, or 3.65 mmol of Nd(NO_3_)_3_·6H_2_O (Neodymium(III) nitrate >99%, Chemcraft Ltd., CAS #16454-60-7), or 4 mmol of Tm(O_2_C_2_H_3_)_3_·4H_2_O (Thulium(III) acetate tetrahydrate >99%, Chemcraft Ltd. CAS #207738-11-2) were processed for 60 min inside the microwave oven at 1350 W to form Ag-, Ce-, Fe-, Ho-, MoS_2_-, Nd- or Tm-GQDs, respectively. The concentrations of silver and iron were calculated using the EDX analysis: 0.0026 mg/mL and 0.0059 mg/mL [80], respectively. N-GQDs were synthesized the same way with a single glucosamine precursor only. Al-GQDs were synthesized by mixing a 1:1 mol ratio of glucose:urea in 10 mL of DI water with 0.0136 mmol of Tris(8-hydroxyquinoline)aluminum(III) (Alq3) and 250 mL of ethanol, after which the mixture was ultrasonically processed for 2 h (Q55, Amplitude 40, QSonica, Newtown, CT, USA) and microwave-treated for 60 min at 1350 W.

#### 2.1.2. Top-Down Approach

R-GQDs were synthesized via a top-down approach using a benchtop UV transilluminator (LMS-20, 8 W). Briefly, 4 mg of RGO (High Porosity Reduced Graphene Oxide, SKU #HP-RGO-025G, Graphene Supermarket, Ronkonkoma, NY, USA) was suspended in 20 mL of DI water with further addition of 1 mL of NaOCl (Sodium hypochlorite 5%, LC246304, LabChem, Zelienople, PA, USA). The suspension was tip sonicated (QSonica, Q55) at 0.3W for 60 min in an ice bath and then UV-irradiated at 302 nm for 2 h to facilitate a photochemical reaction yielding R-GQDs [86].

### 2.2. Purification

In order to remove unreacted precursors in the bottom-up and top-down synthesis, samples are transferred to a 1 kDa molecular weight cutoff (MWCO) bag and dialyzed for 24 h against DI water. The water was changed every 30 min for the first 3 h, after which it was changed every 7 h. For the top-down approach, purification from the unreacted RGO precursor was accomplished by using 0.22 μm syringe filtration. Furthermore, all top-down and bottom-up synthesized GQD materials were further filtered through the 0.22 μm syringe filter to sterilize the product.

### 2.3. Optical Characterization

Photoluminescence spectra of all N-GQDs in the visible region were collected using a Horiba Scientific SPEX NanoLog spectrofluorometer. The absorbance of all samples was measured within the range of 200–700 nm by an Agilent Technologies Cary 60 UV–vis absorption spectrometer. The concentration of GQDs was derived using the Beer–Lambert law (A = εCl). For GQDs synthesized from glucosamine precursor, the extinction coefficient was found to be 0.39 mL mg^−1^ cm^−1^ at 278 nm, while the extinction coefficient of R-GQDs was calculated previously as 3.66 mL mg^−1^ mm^−1^ at 230 nm [86]. Fluorescence spectra were measured at 1 mg/mL for all GQDs. Absorption spectra for each GQD were calculated for the samples at the following concentrations: Ag- (1.77 mg/mL), Al- (1.01 mg/mL), Ce- (2.15 mg/mL), Fe- (1.90 mg/mL), MoS_2_- (1.68 mg/mL), N- (0.76 mg/mL), Ho- (1.01 mg/mL), Nd- (0.80 mg/mL), Tm- (2.02 mg/mL), R- (4.05 mg/mL). Fluorescence spectra for all GQDs were measured at 1 mg/mL.

## 3. Methods

### 3.1. Cell Culture

Human embryonic kidney 293 (HEK-293) cells were utilized for in vitro experiments. The cells were cultivated in Dulbecco’s modified Eagle medium (D6046, Sigma-Aldrich, St. Louis, MO, USA) and supplemented with 10% fetal bovine serum (16140-063, Gibco, Billings, MT, USA), l-glutamine (G7513, Sigma-Aldrich, St. Louis, MO, USA), minimum essential medium non-essential amino acid solution (M7145, Sigma-Aldrich, St. Louis, MO, USA), and 1% penicillin/streptomycin (P4333, Sigma-Aldrich, St. Louis, MO, USA). The cell culture was placed in a Midi 40 CO_2_ incubator (3403, Thermo Scientific, Waltham, MA, USA) at 37 °C with 5% CO_2_. Subsequently, the cultured cells were utilized for both the cell viability assay and cell imaging using fluorescence microscopy.

### 3.2. MTT Assay

The MTT assay was utilized to assess the biocompatibility of the doped GQDs. MTT, standing for 3-(4,5-dimethylthiazol-2-yl)-2,5-diphenyltetrazolium bromide, is a yellow tetrazolium salt converted into a purple formazan product by mitochondrial enzymes in metabolically active cells. In the present work, HEK-293 cells were seeded into 96-well plates (5000 cells per well) for approximately 18 h. Next, they were pre-treated with increasing concentrations of the GQDs and placed in the 5% CO_2_ incubator. After 16 h, the media supernatant was replaced by 3-(3,4-dimethylthiazole-2-yl)-2,5-diphenyltetrazolium bromide (MTT, Sigma) at 1 mg/mL with serum-free medium and incubated for an additional 4 h. The MTT solution was then replaced by a medium with dimethyl sulfoxide (DMSO), and the plates were shaken at room temperature for 5 min. The absorbance of the plates was then analyzed at 540 nm to evaluate cell viability [87].

### 3.3. Fluorescence Microscopy Imaging

Fluorescence microscopy studies were performed with an Olympus IX73 fluorescence microscope equipped with a 60× (IR-corrected Olympus Plan Apo) water immersion objective. The microscope was coupled to a Photometrics Prime 95B CMOS camera through an Olympus DSU confocal system for confocal detection of intracellular GQD fluorescence in the visible spectrum. The 480 ± 20 nm excitation and 535 ± 20 nm emission filters were chosen in accordance with the GQD excitation and emission spectra.

### 3.4. Slide Preparation

In total, 10,000 cells were seeded onto sterilized coverslips in a 6-well plate. These coverslips were coated with rat tail collagen I (ALX-522-435-0020, Enzo, Manhattan, NY, USA) to facilitate HEK-293 cell attachment. Once the cells adhered to the coverslips, metal-doped GQD samples were added to the cell growth medium and incubated for 12 h. Subsequently, the coverslips containing cells were washed with 1× phosphate-buffered saline (PBS) to eliminate any non-internalized GQDs. The cells were then fixed using a 4% formaldehyde solution (28908, Thermo Scientific) at 4 °C for 30 min, and 1× Fluoromount-GTM mounting medium (00-4958-02, Invitrogen, Waltham, MT, USA) was used before being sealed onto microscope slides.

### 3.5. Non-Cellular Antioxidant Assays

**DPPH Assay.** (2,2-diphenyl-1-picrylhydrazyl radical.) The DPPH radical scavenging assay was used to test the antioxidant properties of the GQDs. A total of 100 μM of DPPH• radical (DPPH• free radical 95%, Lot #Z29G013, Alfa Aesa, Ward Hill, MT, USA) was dissolved in 1 mL of ethyl alcohol. After mixing, the solution was incubated in the dark for 1 h. After that, 1 mL of each doped GQD sample at 1 mg/mL was mixed with the free radical solution. A color change was observed if antioxidants were present. Thus, the radical scavenging effectiveness was quantified through an absorption decrease measured at 520 nm with a Bio-Tek Instruments UQuant Microplate Spectrophotometer (71777-1, Bio-Tek, Winooski, VT, USA).

**KMnO_4_ Assay.** The potassium permanganate assay was also used as a reduction test to evaluate the antioxidant properties of the GQDs. A total of 2 mL of acidified (pH = 3) KMnO_4_ (Potassium permanganate ACS reagent >99%, Lot #MKBW7980V, Sigma-Aldrich, St. Louis, MO, USA) at 100 mM was mixed with 1 mg of each respective doped GQDs, either solid or dispersed in water at 1 mg/mL. After incubating the mixture in the dark for 30 min, a reduction of potassium permanganate could be observed. The solution color change from purple to colorless suggested the presence of antioxidant properties of the tested agent. The degree of radical scavenging was quantified by the absorption of KMnO_4_ at 515 nm, measured with a Bio-Tek Instruments UQuant Microplate Spectrophotometer (71777-1, Bio-Tek, Winooski, VT, USA).

**Dye Protectant Assay.** Rhodamine B (RhB) was used as a target molecule for ROS-based oxidation and absorption bleaching in the dye protectant assay. A photochemical reaction was initiated with UV light (LMS-20, 8 W) using hydrogen peroxide to create ^•^OH radicals. The presence of GQDs with antioxidant properties was expected to protect Rhodamine B dye from being oxidized. To test this, 2 mL aliquots of 25 mM PBS, H_2_O_2_ at 50 μg/mL (Hydrogen peroxide 30–31%, product #95302, Fluka, Morris Plains, NJ, USA), 10 μM of RhB (Rhodamine B, Lot #181115068, Liberty Scientific, Jersey City, NJ, USA), and 1 mg/mL GQD aqueous suspension were mixed for 3 h. The concentration of unreacted RhB was further monitored via absorption spectroscopy at 552 nm using a Bio-Tek Instruments UQuant Microplate Spectrophotometer (71777-1, Bio-Tek, Winooski, VT, USA).

## 4. Results and Discussion

Multiple doped GQD structures were synthesized using a microwave-assisted hydrothermal bottom-up approach from glucosamine starting material with a variety of metal dopants introduced during synthesis. GQDs were further purified, characterized, and assessed for antioxidant properties in this work. Graphitic structures of the doped GQDs were verified through transmission electron microscopy analysis (TEM) for Ag-, Al-, Ho-, N-, and R-GQDs, while the same was done for Ce-, Fe-, MoS_2_- Nd-, and Tm-GQDs in our previous work [80]. All the GQD types used here demonstrate the presence of ordered graphitic lattices (Figure 1). Ag-, N-, and R-GQDs have interplanar distances of 0.26–0.29 nm, likely corresponding to the (111) plane of graphene. Al-GQDs have an interplanar spacing of 0.20 nm, likely corresponding to the (100) plane of graphene, and Ho-GQDs exhibit a spacing of 0.31 nm, likely corresponding to the (002) plane of graphene (Figure 1). To demonstrate the success of metal doping, we performed concomitant EDX measurements assessing the elements present (Appendix A). EDX analysis revealed that Al-GQDs, Ag-GQDs, and Ho-GQDs have metal percentages ranging from 0.26–1.64% (Appendix A). Additionally, our previous works suggest that the metal percentages of Nd-GQDs and Tm-GQDs range from 0.48–0.61% [86], and the metal percentages of Ce-GQDs, MoS_2_-GQDs, and Fe-GQD range from 0.01–0.24 [80].

TEM imaging was further utilized to assess size distributions of Ag-, Al-, Ho-, N-, and R-GQDs with mean diameters ranging from 2.79 ± 0.53 nm for Al-GQDs up to 27.24 ± 5.33 nm for Ag-GQDs. Sizes of other GQD types were previously analyzed in [80]. None of the GQDs used in this study exceed an average size of 50 nm (Appendix A), suggesting that they can be internalized by cells through clathrin-mediated endocytosis [79] without major disruption of cell structure and, thus, maintain substantial biocompatibility.

Absorbance and fluorescence spectroscopies were further utilized to evaluate the optical properties of the synthesized metal-doped and undoped graphene quantum dots. All studied GQDs demonstrate absorption spectra characteristic of functionalized graphitic platform (Figure 2A), including π–π* transitions of aromatic C=C bonds within the GQD graphitic structure at ~200 nm [88], n–π* electronic transitions of the C=O bonds typically observed at ~283 nm [88], and a shoulder arising from π–π* transitions attributed to the C=N groups, usually present at ~317 nm [89]. The variations in peak positions are attributed to the effects of electropositive metal dopants. All doped GQDs display fluorescence in the visible spectrum (Figure 2B), similarly to their undoped counterparts [90]. This suggests that the doping process did not significantly alter GQD fluorescence spectral shape in the visible region. Such behavior is anticipated, as confinement-originated GQD visible fluorescence is predominantly influenced by the size of the GQD graphitic islands encapsulated by the functional groups abundant at the edges [90].

The cytotoxicity of the synthesized GQDs was assessed via an MTT colorimetric assay utilizing a standard MTT reagent (5-diphenyl tetrazolium bromide) in HEK-293 cells. In this assay, all GQDs except Ho-GQDs render over 80% cell viability at over 1 mg/mL, while the latter result in over 80% cell viability at 0.5 mg/mL (Figure 3). Some GQDs, including Ce- and N-doped structures, facilitate over 80% cell viability at concentrations as high as 3 mg/mL (Figure 3). This verifies that the GQDs developed in this work are highly biocompatible compared to many other nanomaterials, including carbon nanotubes and graphene oxide [91,92], and can be used for radical scavenging as well as imaging in vitro [93]. It also sets a general threshold for biocompatible concentrations to be utilized in further studies.

To evaluate the potential bioimaging modality of doped GQDs, their fluorescence was imaged via confocal fluorescence microscopy as they were internalized into HEK-293 cells. The fluorescence images of the control cells without GQD treatment were collected, and the integration time/excitation intensity was adjusted to bring down the intensity of autofluorescence in control samples to a level indistinguishable from the background (Appendix A). At these exact settings, further GQD fluorescence imaging has been performed, thus eliminating cellular autofluorescence from all further images. Imaging results demonstrate that GQDs enter cell cytoplasm, as evidenced by their intrinsic green fluorescence observed within cells at 12 h post-administration (Figure 4 and Appendix A). While cerium- and molybdenum disulfide-doped GQDs exhibited lower emission intensities, they were still distributed evenly throughout the cell’s cytoplasm. To ensure that the GQD’s fluorescence originated from the cell’s cytoplasm, the internalization of N-GQDs is further shown in confocal z-stack images (Appendix A). Since doping does not affect internalization, these images show a representative picture of intercellular GQD translocation. The 12 h imaging time point was chosen, as it provided maximum accumulation of parent undoped structures [94], while a few percent metal doping was not expected to affect internalization substantially.

The DPPH assay is a simple and accurate method to evaluate radical scavenging capabilities: when dissolved in ethanol, DPPH• reflects a deep violet color and absorbs in the 520 nm range. Upon receiving a hydrogen atom, the reduced form (hydrazine) changes its color to a faint yellow. Ascorbic acid, a known antioxidant, was used as a control to verify the accuracy of the assay and spectrophotometric analysis (Figure 5A). R-GQDs are undoped and, thus, were used as a negative control to ascertain the effect of doping. Relative to the other doped GQDs, Al-GQDs showed the highest antioxidant properties, being closest to the positive control at roughly 85% radical scavenging ability (RSA), 50% higher than that of the undoped R-GQDs (Figure 5A). This aligns with previous studies showing that Aluminum exhibits antioxidant properties in mouse brain membranes and microalgae [73,95].

KMnO_4_ is also used to assess the antioxidant properties of the doped GQDs. A decrease in absorbance at 515 nm can be observed after Mn^7+^ is reduced by antioxidants to Mn^2+^ in an acidic KMnO_4_ solution. The reducing capacity of the GQDs was therefore estimated by directly comparing absorption in solution to that of a control solution treated with ascorbic acid. Aluminum-, thulium-, and cerium-doped GQDs all exhibited substantial antioxidant activity by yielding ~60% absorbance decrease (60% increase in radical scavenging activity) over that of the undoped GQDs (Figure 5B), while the other GQDs showed less drastic changes. It is noteworthy that in both KMnO_4_ and DPPH assays, Al-GQDs showed consistent reducing potential. All the doped nanomaterials, however, performed above the undoped R-GQDs, suggesting again that doping increases antioxidant properties against various radicals (Figure 5B).

In the third assay, the GQDs were tested for protecting Rhodamine B dye from degradation by hydroxyl radicals to further confirm the radical scavenging abilities of various doped GQD structures. A photochemical reaction was initiated with hydrogen peroxide to create hydroxyl radicals enabling dye oxidation. The dye changes colors from magenta to clear as it oxidizes if no antioxidants are present to offset the reaction [79]. Thus, the ratio of remaining dye quantified by the absorption spectra and compared to positive (ascorbic acid) and negative (R-GQDs) controls (Figure 5C) served as a measure of the RSA. In this experiment, Al-GQDs maintained their strong antioxidant performance with a high dye protection efficiency, while Fe-GQDs also showed similar properties. All other GQDs also showed substantial oxidation protection within 10–30% of the positive control and ~30–50% above R-GQDs (Figure 5C).

The graphene quantum dots (GQDs) used in this work are composed of a sp^2^ hexagonal carbon lattice featuring various oxygen-based functional groups. When highly electropositive elements such as alkali metals are incorporated into the GQDs’ structure through heteroatom doping, the result is an increase in negative electron densities on the adjacent carbon atoms. This effect is due to the electropositive elements’ inclination to donate electrons, which consequently assigns a partial negative charge to neighboring carbon atoms [79]. As a result, the doping process is anticipated to amplify the antioxidant properties of GQDs. This enhancement is largely due to the generation of regions with altered electron density within the graphene structure of the carbon lattice, which augments the interactions between the free radicals and the GQDs. The irregular charge distribution resulting from this process generates a degree of polarity, while the expansive surface area of the GQD platform acts as an electron sink. Collectively, these characteristics could potentially heighten the radical scavenging ability of the GQDs.

As a result of all three assays, Al-doped GQDs appear to exhibit superior antioxidant properties by consistently surpassing R-GQDs serving as a negative control for GQD doping by over 50% in antioxidant activity and being within ~10% of the positive control for all assays (Figure 5). Tm- and Ce-doped GQDs also showed substantial radical scavenging abilities within 5% of the positive control at ~95% RSA in the KMnO_4_ assay (Figure 5B). It has been previously demonstrated that Aluminum may exhibit antioxidant properties in mouse brain membranes [73], which poses Al-GQDs as potential candidates for therapeutics of ROS-induced brain conditions. Furthermore, Ag, Fe, Ho, Mo, N, and Nd dopants all enhance the reducing properties of the undoped GQDs platform. The undoped GQDs (R-GQDs) averaged at roughly 30% RSA, the lowest of all GQDs, thus indicating that doping increases the reduction power of these nanostructures. Having this similar trend in all three assays verifies the validity of the observed results.

The discrepancies between the assays can be attributed to the specific nature of each. For instance, DPPH• is a relatively stable free radical in comparison to its congeners. The oxidation potential of the DPPH• radical is 0.56 V, while those of hydroxyl and permanganate radicals are 2.80 V and 1.68 V, respectively. Unlike DPPH•, hydroxyl radicals utilized in the RHB assay and originating from peroxide are highly unstable due to an unpaired electron spin configuration, despite neutral electric charges, while potassium permanganate is also unstable due to a high Mn oxidation state. This suggests that DPPH•, being less reactive, is reduced by the GQDs less on average. This can explain why the RSA % for the majority of GQDs in Figure 5A is lower compared to the other assays. The understanding of these inherent differences in reactivity among the various radicals used in assays is crucial, as it underscores the importance of selecting the appropriate assay when evaluating the antioxidant capacity of substances such as GQDs.

Another noteworthy observation is the somewhat lowered RSA percentage of all the samples, including ascorbic acid positive control in the dye protectant assay (Figure 5C). This can be explained primarily by the lag time that it takes ascorbic acid and the GQDs to capture free radicals. In the DPPH and KMnO_4_ colorimetric assays, the color of the solution clears when free radicals are reduced, but in the dye protectant assay, conversely, the dye is degraded by the hydroxyl radicals. Some dye molecules, regardless, will be lost during the process because they are temporarily exposed to hydroxyl radicals before the antioxidants can reach them all. This explains lowered RSA recorded even for the ascorbic acid positive control (Figure 5C). This understanding is crucial, as it allows for a more accurate assessment of the antioxidant strength of the GQDs in the dye protectant assay. Despite lower percentages, we recognize that external factors played a role in the assay.

The observed antioxidant properties make GQDs applicable in a variety of important applications, including the treatment of diabetes [1], rheumatoid arthritis [8,9], cancer [10], and Alzheimer’s disease [2,3,4,5,6,7]. Their high, generally over 1 mg/mL, biocompatibilities allow for using substantial therapeutic doses of these nanomaterials. The location of the GQDs within the cells and biological tissues can be traced via their intrinsic fluorescence to ensure that the antioxidants reach their target. Finally, the large GQD platform offers a capability of concomitant drug delivery that, coupled with their antioxidant properties, can aid in addressing the aforementioned conditions [60]. With all doped GQDs performing at least 30% better than the undoped cores, it is evident that metal and even nitrogen doping can render these nanomaterials more effective in combating the effects of free radicals commonly found in biological systems. Further exploration of the optimal doping conditions and mechanisms underlying GQDs’ radical scavenging ability is essential for harnessing their full potential.

## 5. Conclusions

In this work, we have synthesized ten types of novel doped graphene quantum dots with the aim of generating antioxidants from the doped structures and assessing their radical scavenging abilities with respect to ascorbic acid positive control. All GQDs showed high in vitro biocompatibility, rendering over 80% cell viability at concentrations ranging from 0.5 to 3.5 mg/mL in the MTT assay. The antioxidant properties of the doped structures were evaluated via three common radical scavenging assays: DPPH, KMnO_4_, and Rhodamine B dye protectant assay. Among all nanostructures, Al-GQDs showed the highest reducing power in all three assays by consistently performing within 5–20% of the ascorbic acid positive control. Cerium- and thulium-doped GQDs proved effective at reducing KMnO_4_, while iron-doped GQDs have efficiently protected Rhodamine B against hydroxyl radicals. All other GQDs showed modest success in scavenging the free radicals (~30–60% RSA), which suggests that the reduction ability of synthesized nanomaterials depends strongly on the dopants. The undoped R-GQDs performed the lowest in all three assays, supporting this suggestion. In addition to antioxidant properties, all studied GQDs exhibit substantial fluorescence in the visible spectrum and can effectively internalize into biological cells acting as fluorescence biomarkers. Their high biocompatibility, as well as antioxidant abilities explored for the first time in this work, allow the use of doped GQDs in a plethora of clinical applications requiring radical scavenging. Those may include antioxidant therapeutics for such conditions as cancer, Alzheimer’s, Parkinson’s, and cardiovascular diseases, which can be enhanced by GQD-driven drug or gene delivery.

## Figures and Tables

**Figure 1 antioxidants-12-01536-f001:**
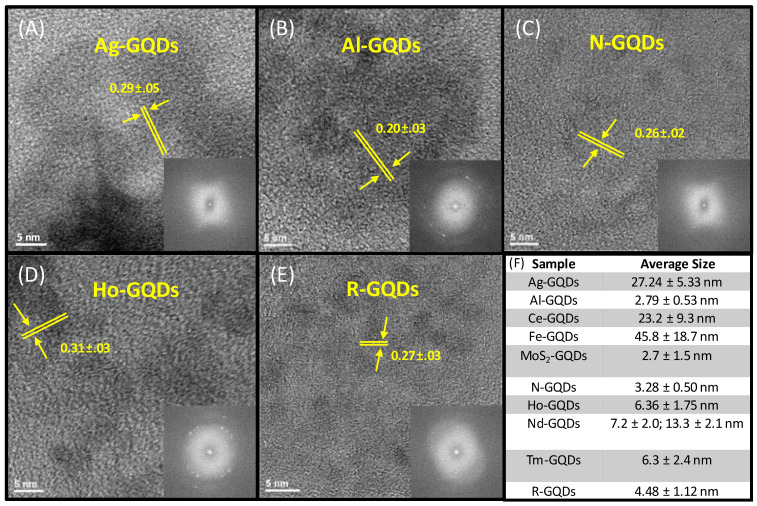
HRTEM images of (**A**) Ag-, (**B**) Al-, (**C**) N-, (**D**) Ho-, and (**E**) R-GQDs, including the corresponding interplanar distances. Inset: FTT images of doped GQDs. (**F**) Average sizes of Ag-, Al-, Ce- [80], Fe- [80], MoS_2_- [80], Ho-, N-, Nd- [80], Tm- [80], or R- GQDs.

**Figure 2 antioxidants-12-01536-f002:**
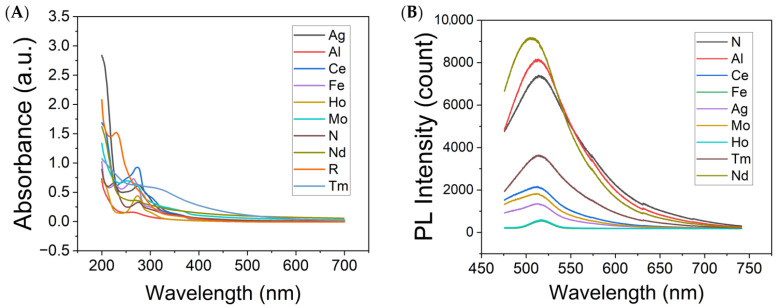
(**A**) Absorption spectra of N-, Al-, Ce-, Fe-, Ag-, MoS_2_-, Ho-, R-, Tm-, and Nd-GQDs. (**B**) Fluorescence spectra of N-, Al-, Ce-, Fe-, Ag-, MoS_2_-, Ho-, Tm-, and Nd-GQDs collected with 460 nm excitation.

**Figure 3 antioxidants-12-01536-f003:**
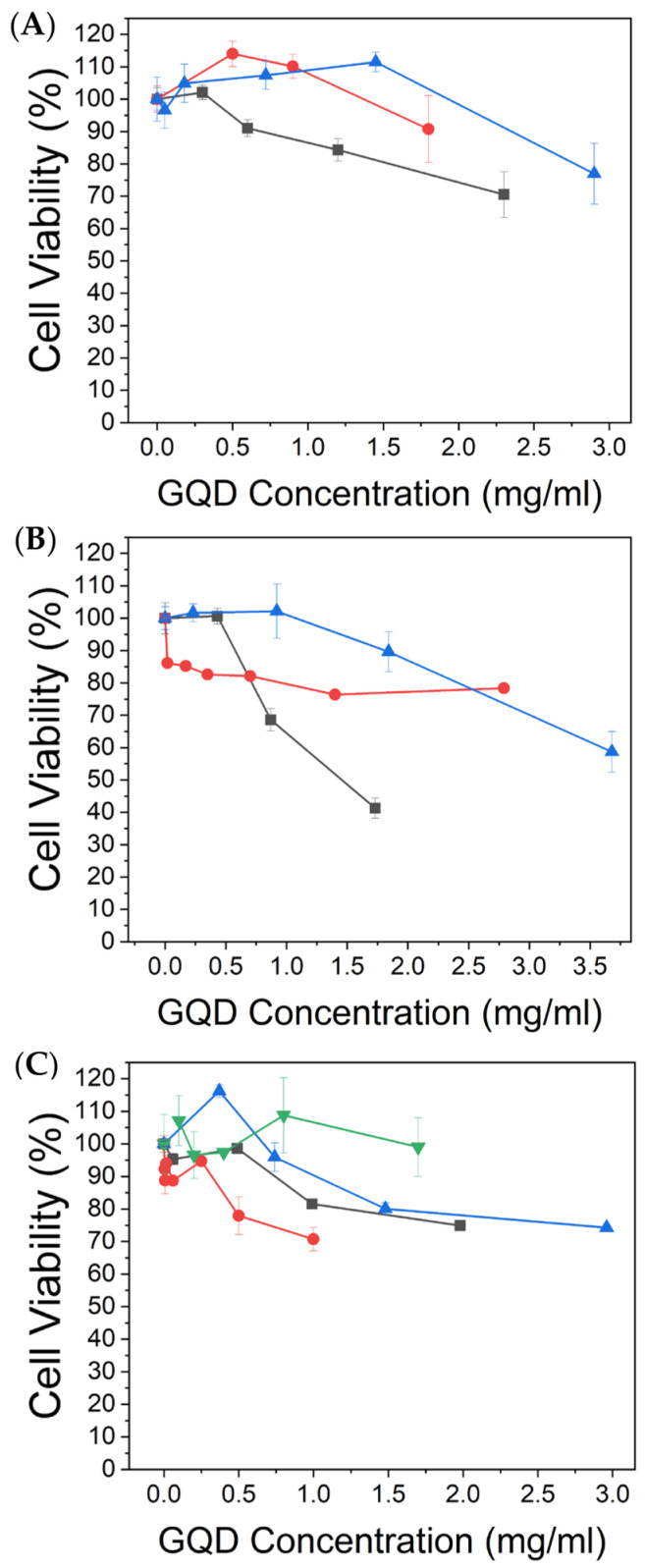
MTT assay: percent HEK-293 cell viability achieved with various concentrations of (**A**) Tm- (gray squares), Ag- (red circles), and N-GQDs (blue triangles). (**B**) R- (gray squares), Al- (red circles), and Ce-GQDs (blue triangles). (**C**) Nd- (blue triangles), Ho- (red circles), Fe- (gray squares), and MoS_2_-GQDs (green triangles).

**Figure 4 antioxidants-12-01536-f004:**
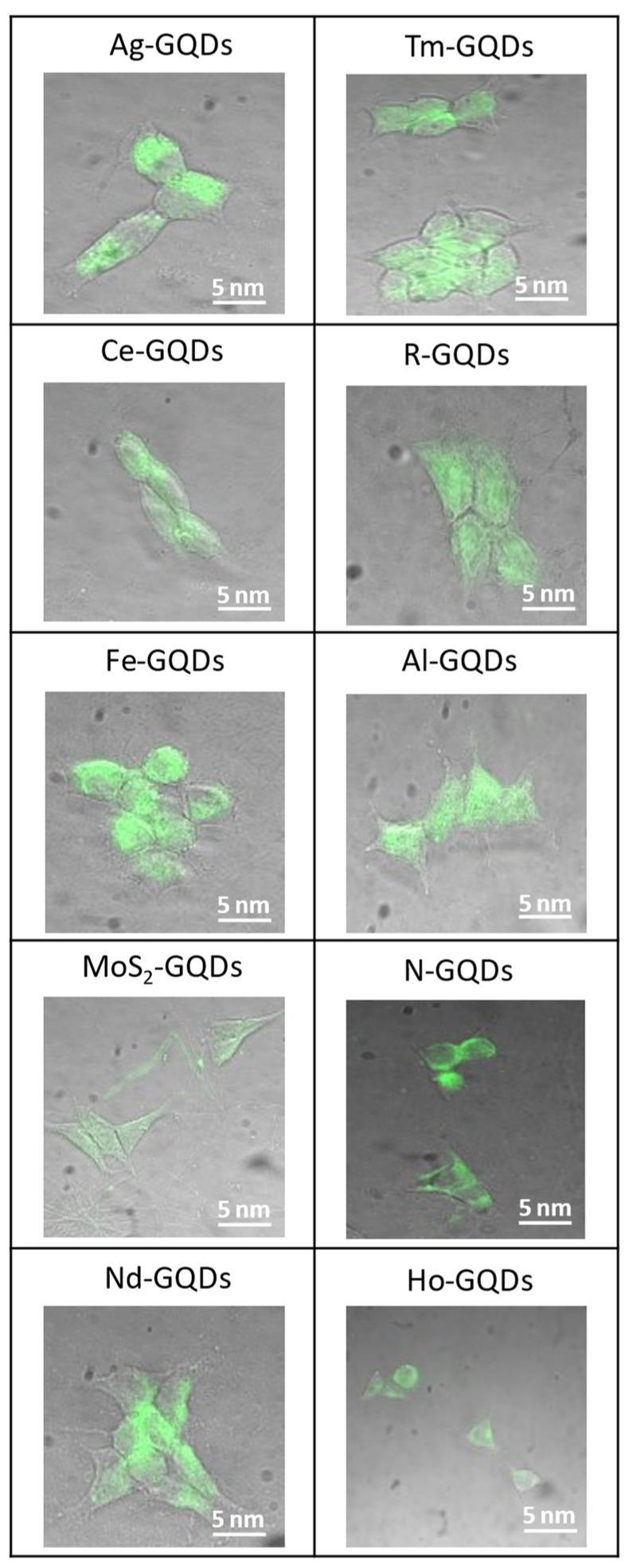
Brightfield/visible fluorescence confocal overlay images of Ag-, Al-, Ce-, Fe-, MoS_2_-, Ho-, N-, Nd-, Tm-, and R-GQDs internalized in HEK-293 cells for 12 h.

**Figure 5 antioxidants-12-01536-f005:**
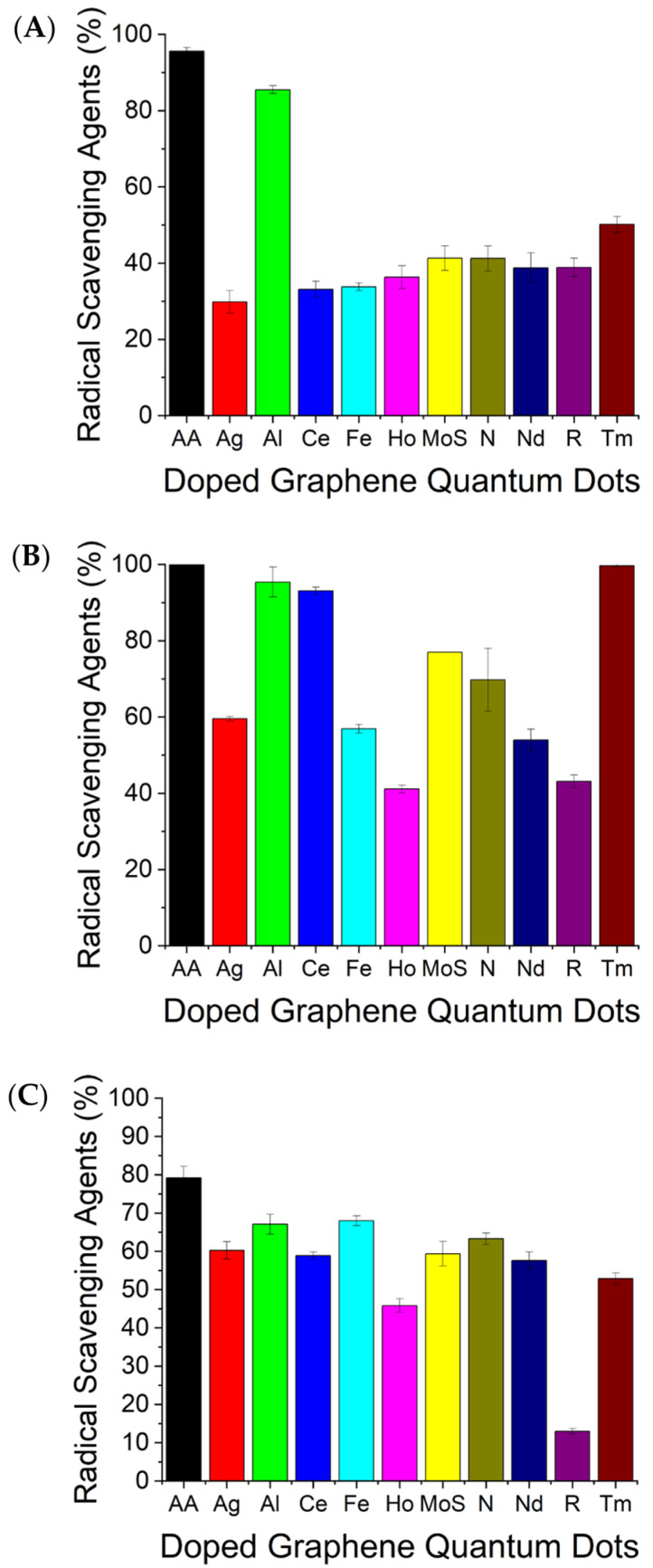
Antioxidant DPPH assays showing the percentage of radical scavenging with respect to ascorbic acid [96] control [96] for Ag-, Al-, Ce-, Fe-, MoS_2_-, Ho-, N-, Nd-, Tm-, or R-GQDs in the (**A**) DPPH radical scavenging assay, (**B**) KMnO4 reduction assay, and (**C**) dye protectant assay.

## Data Availability

The data presented in this study are available on request from the corresponding author. The data are not publicly available due to privacy.

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
