# Peer review of "Doped Graphene Quantum Dots as Biocompatible Radical Scavenging Agents"

_antioxidants, 2023, doi:10.3390/antiox12081536_

Round 1
Reviewer 1 Report
The manuscript by Bhaloo et all deals with antioxidant properties of GQDs, doped with various metal ions. This is potentially important subject, worth investigation. However, there is several points which should be addressed prior to acceptance.
My main problem is experiment with cells. There is no control shown, therefore is impossible to accept, that the emission is of GQD origin only. The green region of fluorescence spectra is prone to show autofluorescence, with maxima e.g. at 510-530 nm (check multiple example in literature). Authors should also check the emission spectra, not only control without GQDs. It is very likely, that some stress my rise the level of cell autofluorescene, even without GQs entering cell interior.
To prove, that GQDs are inside cytoplasm, the authors should definitely show Z-stack images. With one plane image only, the signal might be of nanoparticles adsorbed on the plasma membrane.
Other questions:
How were GQDs purified from synthesis? How was quantified their concentration in further experiments?
Please, explain why the particular cell line was chosen for experiments?
MTT assay: was there a control without cell present, to establish an interaction between GQDs and MTT?
Line 231: It is hard to say, that the absorption characteristics are “similar”. In my opinion, spectra shapes are quite different. The features are explained further, but I would suggest different world choice than “similar”. The same is line 237 “This suggests that the doping process did not significantly alter their optical properties in the visible region”. Right now, I would say there is a difference in bands particular position, as well as extinction coefficient. Additionally, what about fluorescence yield? This looks pretty different, may the authors discuss this point?
Figure 4: the shape of cells seems to vary between types of treatment. Can author comment on that?
Discussion seems to need more references, or reformulation. In some points it is hard to distinguish between authors hypothesis and the “literature fact”. The part of GQDs application as delivery platform is problematic, since a stability of GQDs in the medium was not checked.
Minor points
Figure2: What was the concentration for fluorescence emission Absorption spectra should be zoomed or some insets should be introduced, for easier analysis.
Figure 3: what are the error bars? The quality of this figure is poor, it looks like scanned or copy-pasted without white background maybe.
Figure 4: please, provide separate channels. Overlay images are not perfect for analysis of fluorescence quality.
There is a “Results and Discussion” section, and then “Discussion” again. There are some leftovers of the MDPI templates (see "author contribution section").
Reviewer 2 Report
The paper of Bhaloo et al., entitled ˜Doped Graphene Quantum Dots as Biocompatible Radical Scavenging Agents˜presents the preparation, characterization and antioxidant capacity of several doped graphene quantum dots.
The introduction has to be improved a lot.
page 1, line 37, ˜as a consequence of cellular redox˜ has to be removed because it has no sense.
ROS do not generate nitration. This is an important error. In the first stage, ROS extract a proton. They do not hydroxylate all biomolecules mentioned by authors. Hydroxylation of hydrophobic xenobiotics is a form of detoxification catalyzed by CYPs and in this process ROS are generated. But authors did not explain.
Also, in cells beyond the non-enzymatic antioxidants, an enzymatic system occurs and a discussion regarding this is necessary.
Moreover, Alzeiheimer disease and cancer are developed as a consequence of different levels of oxidative stress and a discussion regarding this is necessary.
Also, the in vitro assays has to be presented in two groups: the non-cellular ones and the cellular ones. MTT assay highlights the mitochondrial activity of cell exposed to the doped graphenes and as a consequence their viability. In my opinion the release of LDH in the cellular media is also necessary in oder to test the cytotoxicity of these compounds.
Reviewer 3 Report
In this work, the authors synthesized ten types of biocompatible graphene quantum dots (GQDs) augmented with a variety of metal dopants. The radical scavenging abilities of these novel structures were for the first time assessed via the DPPH, KMnO4, and RHB (Rhodamine B protectant) assays. While all metal-doped GQDs consistently demonstrate antioxidant properties higher than the undoped cores. This work is interesting, however, there are many issues need to be solved before I can suggest its publication.
1) In the Abstract section, the authors stated that “The radical scavenging abilities of these novel structures were for the first time assessed via the DPPH, KMnO4, and RHB (Rhodamine B protectant) assays”. However, this is not true, as the antioxidant activity of GQDs had been evaluated with DPPH assay, KMnO4 reduction assay, and dye protection assay. (ACS Nano 2016, 10, 8690−8699; Phys. Chem. Chem. Phys., 2017,19, 11631-11638; ACS Appl. Mater. Interfaces 2019, 11, 24, 21822–21829; Carbon, 2017, 116, 366-374).
2) “While all metal-doped GQDs consistently demonstrate antioxidant properties higher than the undoped cores, thulium and aluminum-doped GQDs exhibit up to 60% improvement in radical scavenging compared to ascorbic acid positive control.” I think the authors should clarify this statement, for example, the result corresponding to which assay, DPPH, KMnO4, or dye protection assay, as discrepancies are existed between the three assays.
3) When talking about carbon nanodots applied for scavenging free radicals, some recently published works should be cited (Arab. J. Chem., 2023, 16, 105036; ACS Appl. Mater. Interfaces, 2020, 12, 41088-41095).
4) For the readers to easily repeat the work, the concentration of the silver nanoparticles and iron oxide nanoparticles should be provided.
5) The authors should check the whole manuscript carefully, as there as many errors in the manuscript:
“Tm(O2C2H3)3⋅4H2O” should be “Tm(O2C2H3)3⋅4H2O”; “ml” should be “mL”; Mo-GQD or MoS2-GQDs, should be unified; “Unlike DPPH•, Hydroxyl radicals”; full name of SDS should be given the first time they appeared.
6) The resolution of the TEM images are too low, a better image with high resolution need to be added.
7) From Line 254-255, the authors stated that “Imaging results demonstrate that GQDs enter cell cytoplasm, as evidenced by their intrinsic green fluorescence observed within cells at 12 h post-administration”. However, in the caption of Figure 4, an incubation time of 6 hours was used. The authors should check the time.
8) Relative to the other doped GQDs, Al-GQDs showed the highest antioxidant properties. Why?
There are some typing errors,the authors should check the manuscript carefully.
Round 2
Reviewer 1 Report
I'm satisfied with most of the improvements, introduced to the manuscripts.
However, my two questions still remain unanswered.
First (most probably skipped by mistake, because it was in the row with other answered questions): what about fluorescence yield of all quantum dots? The spectra suggest, it is different. Please, comment on that (even without precise estimation of the value) and propose some explanation for observed effect.
Second: I cannot agree that showing only one plane of the call (even if this is a midplane) proves even cytoplasm distribution. Please, provide Z-stacks.
Reviewer 2 Report
Accept
Author Response
.
Reviewer 3 Report
The manuscript can be accepted now.
Author Response
.